# Actra: Optimized Transformer Architecture for Vision-Language-Action Models in Robot Learning

## Abstract

Vision-language-action models have gained significant attention for their ability to model trajectories in robot learning. However, most existing models rely on Transformer models with vanilla causal attention, which we find suboptimal for processing segmented multi-modal sequences. Additionally, the autoregressive generation approach falls short in generating multi-dimensional actions. In this paper, we introduce Actra, an optimized Transformer architecture featuring trajectory attention and learnable action queries, designed to efficiently process segmented multi-modal trajectories in language-conditioned robot imitation learning. Furthermore, we propose a contrastive dynamics learning objective to enhance its understanding of environment dynamics and multi-modal alignment, complementing the primary behavior cloning objective. Through extensive experiments on three large-scale robot manipulation benchmarks, Actra exhibits substantial performance improvements over state-of-the-art models.

## 1 Introduction

Vision-language-action models (VLAs) have emerged as integral components of recent developments in robot learning. Previous multi-modality models, exemplified by vision-language models (VLMs), have demonstrated proficiency in handling both visual and textual inputs, successfully addressing a spectrum of tasks (Chen et al., 2023) such as visual question answering, image captioning, and image retrieval. Distinctively, VLAs extend beyond the capabilities of VLMs by incorporating the ability to execute actions based on multi-modal inputs. This unique capability empowers VLAs to interpret language prompts, visually perceive their environment, and subsequently execute actions to fulfill the specified tasks. The potential applications of VLAs in robotics are not confined to controlled environments in traditional domains like manufacturing. They also prove their suitability for everyday tasks such as room cleaning and cooking (Brohan et al., 2023b), thanks to their dexterity and generalizability.

To accommodate multi-modal inputs, previous Transformer-based VLMs (Vaswani et al., 2017) explored designing special self-attention schemes to better suit the unique properties of different modalities, such as UniLM (Dong et al., 2019), M6 (Lin et al., 2021), BLIP-2 (Li et al., 2023a) and Octo (Octo Model Team et al., 2023). Consider the task of image captioning as an example, causal attention is not the best option to encode images because there is no clear causal relationship among the image patches. Thus, these VLMs allow bidirectional self-attention for the image tokens while maintaining causal attention for the text tokens.

VLAs predominantly build upon the pioneering foundations laid by Decision Transformer (Chen et al., 2021) and Trajectory Transformer (Janner et al., 2021). These two works frame reinforcement learning (RL) policies as sequence modeling problems, leveraging the expressive power of Transformer models. This paradigm has become a cornerstone across recent VLAs, but both models are Transformer decoders based on causal attention. Subsequent approaches, such as Gato (Reed et al., 2022) and RT-1 (Brohan et al., 2023b), also adopt Transformer decoders as their network backbone, passing in different modalities as a single sequence. VIMA (Jiang et al., 2022) incorporates cross-attention mechanisms to condition the policy with multi-modal prompts, but the decoder stack still follows previous methods and uses causal attention.

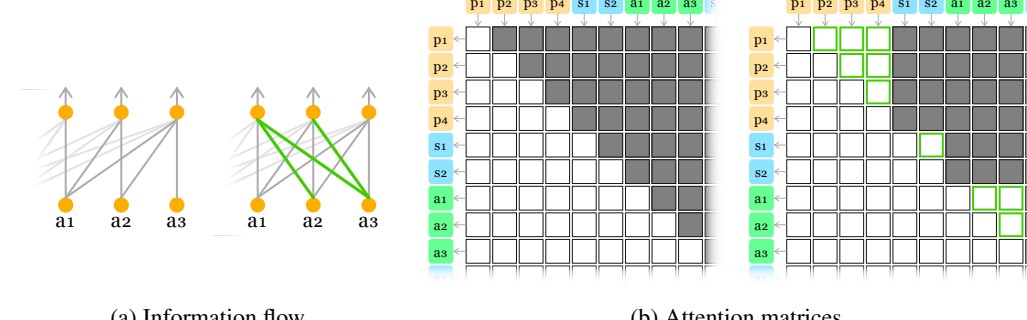

(a) Information flow.      (b) Attention matrices.

Figure 1: (a) Comparison of information flow in causal attention (left) and trajectory attention (right). Each orange dot corresponds to an action dimension. The three action dimensions collectively form an action "segment". Each line represents an attention connection, facilitating information flow from the input token (bottom) to the output token (top). In trajectory attention, tokens can attend to not only the preceding tokens but also the subsequent tokens within the same segment, as indicated by the green lines. (b) Attention matrices of the two attention types. The dark cells represent masked attention, while the green-bordered cells also highlight the additional information flow enabled by trajectory attention.

On the contrary, we have identified that multi-modal trajectories in robotics exhibit unique properties better captured by a novel type of Transformer self-attention, as illustrated in Figure 1 & 2. Specifically, each language prompt, state, or action within a multi-modal trajectory can consist of multiple tokens, referred to as a "segment" in this paper. For instance, robot systems usually make use of several cameras, and as a result, a state is represented with a segment of tokens, each corresponding to a camera. Similar to state tokens, tokens for action dimensions also lack causal relationships with each other. Traditional causal attention hinders full information flow within a segment, as tokens are restricted from attending to the subsequent tokens. To overcome this limitation, we introduce trajectory attention, optimized for multi-modal trajectories. Trajectory attention possesses two key characteristics: inter-segment attention is causal, and intra-segment attention is bidirectional. Since a VLA model only needs to encode the language prompt and follow the corresponding instruction, causal attention is also unnecessary for the prompt segment. Consequently, we advocate for processing trajectories at the segment level, rather than merely at the token level.

To complement trajectory attention, we devise a segment-level decoding scheme that generates a segment as a whole. Drawing inspiration from DETR's object query (Carion et al., 2020), we propose employing action queries to more effectively extract information for action generation. Concretely, we employ one learnable action query for each action dimension. Each action query aggregates the most relevant information in the trajectory for its corresponding action dimension and generates the most probable value for that dimension. Different action queries can execute this procedure in parallel, facilitating the simultaneous generation of all action dimensions. This represents a substantial acceleration in action generation speed compared to earlier approaches that generate one action dimension at a time, such as RT-2 (Brohan et al., 2023a). By combining trajectory attention and action queries, we introduce an optimized Transformer architecture for multi-modal trajectories, which we name **Act**ion-query-based **Tra**jectory-attention **Tra**nsformer, or **Actra** for short.

While training the policy network with the behavior cloning objective is a common practice in VLAs, numerous prior approaches have also explored incorporating auxiliary objectives to further improve performance. Dynamics learning methods (Li et al., 2024; Sun et al., 2023; Liu et al., 2022) encourage the model to understand how the environment responds to its actions. These methods typically fall into two categories: forward dynamics prediction and inverse dynamics prediction. Forward dynamics prediction aims to predict the next state given the current action, but it requires additional state decoders, which increases model complexity. Conversely, inverse dynamics prediction seeks to predict the action taken between two given consecutive states. Although conceptually distinct from behavior cloning, the difference is nuanced in practice: inverse dynamics prediction reconstructs actions using a masked modeling strategy, and behavior cloning also predicts actions, albeit in an autoregressive manner.

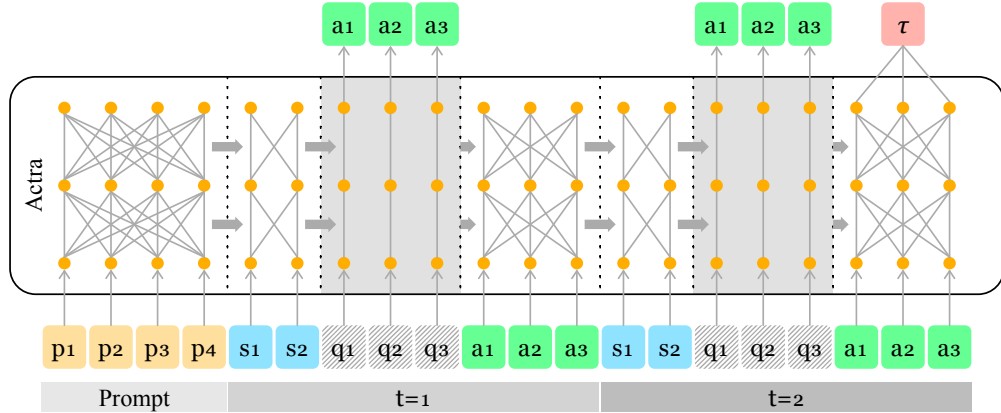

Figure 2: The architecture of Actra. Each square represents a token. A trajectory $\tau$ consists of a prompt segment $p_{1:4}$, state segments $s_{1:2,t}$, action segments $a_{1:3,t}$. For clarity, the trajectory is a simplified example and does not reflect the actual specifications of Actra. In the Transformer, vertical dashed lines divide the segments. Learnable action queries $q_{1:3,t}$ are inserted after each state segment to extract information for action generation. Each token embedding (orange dot) in the trajectory can attend to embeddings from all previous segments (horizontal arrows), as well as all the embeddings in its own segment (gray lines). Notably, action queries, which contain no trajectory information, are hidden from other tokens, but they still attend to all preceding tokens. In addition to its primary function of decoding actions, Actra can also encode the entire trajectory by pooling the embeddings in the last segment (red box).

We propose a novel contrastive dynamics learning (CDL) method, where augmented positive trajectories are contrasted with negative trajectories that mismatch states and actions from different trajectories, as illustrated in Figure 3. During CDL training, the model is encouraged to determine whether a trajectory correctly follows the dynamics of the environment. This capability is crucial for the agent to understand the consequences of its actions and make more informed decisions. The implementation of CDL requires only the addition of a simple classification head, consisting of a pooling layer and a linear layer. Furthermore, CDL also serves as a unified representation learning approach for multi-modal trajectories. To accurately capture environment dynamics, the model relies on effective vision, language, and action encoders to differentiate positive samples from negative ones, leveraging their multi-modal encoding capabilities. Several studies have demonstrated the efficacy of contrastive learning in aligning different modalities (Radford et al., 2021; Jia et al., 2021; Yuan et al., 2021; Yao et al., 2022). Since prompts, states, and actions represent three distinct modalities of VLAs, CDL can also be viewed as a method for enhancing alignment across the vision, language, and action modalities.

The three main contributions of this paper are:

- We introduce Actra, an optimized Transformer architecture featuring trajectory attention and action query, designed to efficiently process multi-modal trajectories on the segment level;

- We propose a contrastive dynamics learning objective to explicitly improve Actra's understanding of environmental dynamics and enhance its multi-modal encoding capabilities, complementing robot imitation learning;

- Extensive experimental results across three large-scale robot manipulation benchmarks demonstrate that Actra significantly outperforms state-of-the-art vision-language-action models, showcasing the effectiveness of our approach.

## 2 RELATED WORK

**Vision-language-action Model.** Vision-language-action models (VLAs) constitute a class of multi-modal models designed to generate actions based on specified language prompts and perceived

environment. Coined by RT-2 (Brohan et al., 2023a), VLAs have garnered increasing attention due to their dexterity and generalizability in handling complex robotics tasks. Early attempts were based on existing vision-language models (VLMs), exemplified by CLIPort (Shridhar et al., 2021) and BC-Z (Jang et al., 2021). Gato (Reed et al., 2022) explored the use of a single Transformer model (Vaswani et al., 2017) as the control policy for tasks spanning various domains, unifying multi-modal inputs into a single sequence. RT-1 (Brohan et al., 2023b) stands as a dedicated robotics transformer for robotics tasks. Our model is also a VLA but with an optimized Transformer architecture.

**Multi-modal Transformer.** Several VLMs, including UniLM (Dong et al., 2019), M6 (Lin et al., 2021), and Octo (Octo Model Team et al., 2023) have endeavored to optimize Transformer's self-attention for vision-language inputs. Despite these efforts, adapting self-attention to the multi-modal inputs of VLAs has been relatively unexplored. Gato (Reed et al., 2022) and RT-1 (Brohan et al., 2023b) maintain causal attention in Transformer decoders. VIMA (Jiang et al., 2022) proposes passing the prompt into the policy through cross-attention, but their Transformer decoder stack still employs causal attention. To the best of our knowledge, Actra is the first VLA designed to accommodate multi-modal trajectories with a unique self-attention mechanism.

First introduced in DETR (Carion et al., 2020), learnable object queries have shown promising results in extracting information for object detection. BLIP-2 (Li et al., 2023a) used a similar strategy to extract visual embeddings for vision-language tasks. In our approach, we employ learnable action queries at the action-dimension level to extract information most relevant to individual action dimensions.

**Dynamics Learning & Multi-modal Contrastive Learning.** Dynamics learning has long been recognized as a powerful technique for improving the performance of robot learning models. Dreamer (Hafner et al., 2020) was a pioneering work in this domain, inspiring several follow-up methods, including Iso-Dream (Pan et al., 2022), TWM (Robine et al., 2023), and IRIS (Micheli et al., 2023). Many recent dynamics learning approaches (Li et al., 2024; Sun et al., 2023; Liu et al., 2022) can be classified into two categories: forward dynamics prediction and inverse dynamics prediction. Most of these methods rely on generative models coupled with additional decoder modules, such as video generators (Du et al., 2023). Our contrastive dynamics learning approach is based on contrastive learning and only involves an encoding process.

A series of VLMs, CLIP (Radford et al., 2021), ALIGN (Jia et al., 2021), Florence (Yuan et al., 2021), FILIP (Yao et al., 2022), has demonstrated the significance of contrastive learning in enhancing multi-modal interaction. However, in VLA models like R3M (Nair et al., 2022) and VIP (Ma et al., 2023), where contrastive learning has been adopted, the primary emphasis remains on improving visual representations. In contrast, our proposed contrastive dynamics learning task explicitly compels the model to align all three modalities—vision, language, and action—thereby enabling more effective encoding of multi-modal inputs.

## 3 OUR METHOD

### 3.1 PRELIMINARIES

Markov Decision Process (MDP) comprise states ($s$) and actions ($a$) and it can be conditioned by a language prompt ($p$). In the context of imitation learning, a multi-modal trajectory within the language-conditioned MDP is denoted as $\tau = (p, s_{t=1}, a_{t=1}, \ldots, s_{t=T}, a_{t=T})$. Each element in the trajectory—$p$, $s_t$, or $a_t$—comprises a segment of tokens. For instance, a state $s_t$ corresponds to the segment $s_{1:M,t} = (s_{1,t}, s_{2,t}, \ldots, s_{M,t})$, where each element is a token. Tokens in $p$ are standard NLP tokens. State tokens in $s_t$ correspond to scene images or object images. Action tokens in $a_t$ contain SE(2) actions or 6D poses. Therefore, a trajectory at the token level is written as $\tau = (p_{1:L}, s_{1:M,t=1}, a_{1:N,t=1}, \ldots, s_{1:M,t=T}, a_{1:N,t=T})$. The goal is to train a policy that can generate an optimal action based on the past trajectory $\pi_\theta(a_t | p, s_{\leq t}, a_{<t})$.

### 3.2 ACTRA

In natural language generation (NLG), language models such as GPT (Radford et al., 2019) employ Transformer decoders as the backbone. To prevent tokens from having visibility into subsequent

tokens, a causal attention mask is applied in the Transformer decoder. Prior VLA models (Chen et al., 2021; Brohan et al., 2023b) have followed this trend for action generation. While causal attention is well-suited for NLG, where language tokens are sequentially generated, it is not the optimal attention mechanism for modeling multi-modal trajectories in robot learning.

**Trajectory attention.** Images of the state $s_t$ from multiple cameras arrive simultaneously, lacking causality among themselves. They are determined solely by the preceding action $a_{t-1}$ and the environment. The same principle applies to actions: $a_t$ is only dependent on previous states and actions, and they are conditionally independent from each other. The action dimensions in an action do not exhibit a clear causal order. For instance, in a 3D coordinate, it is uncertain whether $a_{1,t}$ depends on $a_{2,t}$ or vice versa. Regarding the language prompt, as it is provided by the user, the model's job is to encode and understand it rather than generate the prompt, akin to BERT (Devlin et al., 2019). Vanilla causal attention might impede information flow within each segment of a multi-modal trajectory, prohibiting $s_{1,t}$ from attending to $s_{2:M,t}$, and $s_{2,t}$ from attending to $s_{3:M,t}$, and so forth. This similarly holds for prompts and actions.

To address the issue, we propose an optimized Transformer self-attention mechanism for language-conditioned multi-modal trajectories, termed *trajectory attention*. Trajectory attention exhibits two key properties: the inter-segment connections are causal, and the intra-segment connections are bidirectional. Its corresponding attention matrix is illustrated in Figure 9. Following the convention of the Transformer attention matrix, we designate the row index as the destination of self-attention and the column index as the source. Consequently, the causal attention matrix has all its lower triangle entries, $(i, j)$ for $i \geq j$, set to one, and the rest set to zero. Trajectory attention is achieved by unmasking the entries in the causal attention matrix corresponding to $(p_i, p_j)$, $(s_{i,t}, s_{j,t})$ or $(a_{i,t}, a_{j,t})$ for $i < j$. When compared with the original causal attention, there are $L(L-1)/2 + T\big(M(M-1)/2 + N(N-1)/2\big)$ additional entries joining the self-attention in every Transformer layer, which explains the effectiveness of trajectory attention. As a result, Actra is designed to process multi-modal trajectories at the segment level, which aligns well with the MDP setting as it involves states and actions rather than individual tokens.

**Action query.** Adapting to the segment-level trajectory attention mechanism, we introduce a segment-level decoding scheme based on learnable action queries. Most prior VLAs generate action dimensions autoregressively, where each action dimension depends on its preceding token embedding (Brohan et al., 2023a). However, this approach is suboptimal because the embedding of the preceding token is highly dependent on its input and may lack the most relevant information for the action dimension. For instance, when generating $a_{1,t}$, its preceding token is $s_{M,t}$. Although the embedding of $s_{M,t}$ can aggregate information from the past trajectory through self-attention, it is largely influenced by its corresponding input image and may not contain sufficient information about $a_{1,t}$. To overcome this limitation, we adopt learnable action queries $q_{1:N}$ for individual action dimensions $a_{1:N}$, inspired by DETR (Carion et al., 2020). Each action query $q_i$ is dedicated to one action dimension $a_i$ and is shared across all timesteps: $q_{i,t=1} = q_{i,t=2} = \cdots = q_{i,t=T}$ for $i \in \{1 \ldots N\}$. We argue that this approach can find more relevant information for each action dimension because the action query $q_i$ can exclusively attend to information pertinent to $a_{i,t}$. Since action queries have no associated input token, their embeddings fully retain action dimension information. Moreover, distinct from autoregressive generation, action queries can extract information and generate all dimensions of an action segment in parallel. Therefore, the decoding procedure operates at the segment level. This significantly speeds up action generation. As the action queries are solely used for information extraction and do not hold any trajectory information, they are masked out from the attention matrix, ensuring that other tokens cannot see them through the self-attention mechanism.

**Actra** Combining trajectory attention and action query, we introduce a novel Transformer variant named Actra. In Actra, all action tokens $a_{i,t}$ can fully attend to $(p, s_{t=1}, a_{t=1}, \ldots, s_t, a_t)$, and all state tokens $s_{i,t}$ can fully attend to $(p, s_{t=1}, a_{t=1}, \ldots, s_t)$. Consequently, their embeddings are enhanced for multi-modal trajectories. Each action query $q_{i,t}$ aggregates this enriched information, collecting more pertinent information for its corresponding action dimension. This makes Actra a more suitable Transformer for action generation in multi-modal trajectories. The training process utilizes standard behavioral cloning in robotic imitation learning, optimizing the objective $\mathcal{L}_{\text{BC}} = \min_\theta \sum_{t=1}^{T} -\log \pi_\theta(a_t | p, s_{\leq t}, a_{<t})$ on offline expert trajectories.

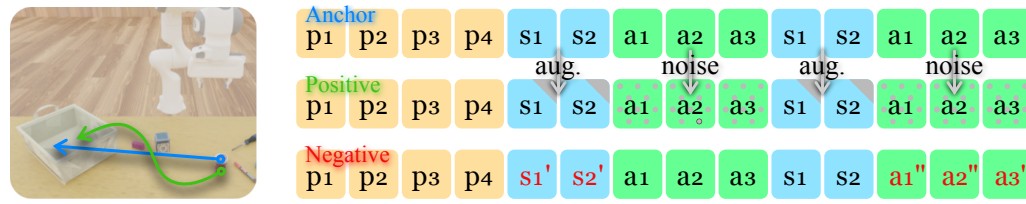

(a) Action perturbation.  (b) Construction of positive and negative samples.

Figure 3: Overview of contrastive dynamics learning. (a) In the anchor trajectory (blue arrow), the object on the right is picked up and placed into the bin on the left. A slightly deviated trajectory (green arrow) can still reach the desired destination, allowing for action perturbation to be used in constructing positive samples. (b) Given the anchor, we construct a positive sample by applying image augmentation and the proposed action perturbation. Negative samples are created by mismatching states and actions from other trajectories.

## 3.3 Contrastive Dynamics Learning

The primary behavior cloning objective is to train a model to predict the next action based on the past trajectory. Dynamics learning encourages the model to learn how the environment transitions from one state to another based on the agent's actions, enabling it to generate more informed decisions. Our contrastive dynamics learning (CDL) objective introduces minimal changes to the model architecture, requiring only an additional classification head composed of a pooling and linear layer. As illustrated in Figure 3b, we construct positive samples by augmenting the anchor trajectory using standard image augmentation and a novel action perturbation technique. Negative samples are created by mismatching states and actions from different trajectories.

Concretely, we assume that the anchor trajectory is $\tau = (p, s_{t=1}, a_{t=1}, \ldots, s_{t=T}, a_{t=T})$. To construct a positive sample, $\tau^+$, we first apply standard computer vision data augmentation techniques to state images, such as random cropping. Additionally, we introduce a novel approach for augmenting actions. The intuition is that a slightly deviated path can still lead the agent to the desired destination, as shown in Figure 3a. To achieve this, we perturb the actions by adding a small amount of random noise. By combining image augmentation and action perturbation, the positive sample is an augmented version of the anchor trajectory.

Subsequently, we create negative trajectories that violate the correct environment dynamics. Given different trajectories from the anchor, $\tau' = (p', s'_{t=1}, a'_{t=1}, \ldots, s'_{t=T}, a'_{t=T})$ and $\tau'' = (p'', s''_{t=1}, a''_{t=1}, \ldots, s''_{t=T}, a''_{t=T})$, we mismatch their states and actions with those of the anchor trajectory to construct negative samples: $\tau^- = (p, s'_{t=1}, a_{t=1}, \ldots, s_{t=T}, a''_{t=T})$. These strong negatives are constructed based on the following principles discovered during the development of CDL. First, we refrain from inserting entirely random actions or states, as these have not appeared in the dataset and can be easily identified as negatives. Second, instead of mismatching only the original states and actions, we also use augmented ones. This prevents models from trivially identifying positive samples by detecting the presence of image augmentation or action perturbation. Third, we avoid merely shuffling states and actions along the time axis, as such negatives are also easily recognizable. The overall CDL objective is to contrast the augmented positive trajectory $\tau^+$ against various negative trajectories $\tau^-$.

In contrastive dynamics learning, Actra encodes the entire multi-modal trajectory into a sequence of embeddings. Due to our trajectory attention mechanism, the action tokens at the final timestep attend to the entire trajectory. Their token embeddings are then aggregated into a single trajectory embedding using a simple classification head, consisting of a pooling layer (Li et al., 2023b) and a linear layer, as shown in Figure 2. We denote this process as $f(\cdot)$. Finally, we employ the standard InfoNCE objective (van den Oord et al., 2018) in contrastive learning to train the model to distinguish between embeddings of positive and negative trajectories:

$$\mathcal{L}_{\text{CDL}}(\tau, \tau^+, \tau^-) = -\log \mathbb{E}\left[\frac{\exp(f(\tau) \cdot f(\tau^+))}{\exp(f(\tau) \cdot f(\tau^+)) + \sum_i \exp(f(\tau) \cdot f(\tau_i^-))}\right]. \tag{1}$$

Table 1: Performance comparison of success rate (%) on the VIMA-Bench benchmark.

| Model | Configuration | | Params | Generalization Levels | | | | Overall |
|---|---|---|---|---|---|---|---|---|
| | Attn Type | Visual Token | | L1 | L2 | L3 | L4 | |
| DT | Self Attn | Single Image | 42.0M | 47.69 | 46.92 | 43.33 | 12.50 | 37.61 |
| Gato | Self Attn | Image Patches | 42.0M | 45.38 | 42.31 | 40.00 | 15.00 | 35.67 |
| Flamingo | Cross Attn | Image Perceiver | 42.4M | 44.62 | 43.85 | 41.67 | 10.00 | 35.04 |
| VIMA | Cross Attn | Object Tokens | 42.4M | 78.85 | 78.46 | 81.67 | 47.50 | 71.62 |
| Actra (ours) | Traj Attn | Object Tokens | 37.8M | 83.08 | 81.54 | **84.00** | **50.00** | **74.66** |
| w/ CDL | Traj Attn | Object Tokens | 37.8M | **86.92** | **86.15** | 83.33 | 35.00 | 72.85 |

# 4 EXPERIMENTAL RESULTS

## 4.1 EXPERIMENTAL SETUP

We compare our approach with various baseline models across three different benchmarks: VIMA-Bench (Jiang et al., 2022), Maniskill2 (Gu et al., 2023), and CALVIN (Mees et al., 2022). Each benchmark emphasizes different aspects of robot learning. VIMA-Bench investigates multi-modal robot learning, where prompts to agents are multi-modal; its evaluation assesses the generalization capacity to novel adjectives, nouns, and even meta-tasks. Maniskill2 targets everyday objects with complex geometries, testing generalization to unseen geometric and visual attributes. CALVIN, on the other hand, examines long-horizon manipulation tasks, assessing how well models generalize to new environments.

We include state-of-the-art VLA models as baselines, including DT (Chen et al., 2021), Gato (Reed et al., 2022), Flamingo (Alayrac et al., 2022; Jiang et al., 2022) and VIMA (Jiang et al., 2022). In Maniskill2, we compare against RT-1 (Brohan et al., 2023b) instead since it addresses the same task. In VIMA-Bench and CALVIN, Actra (38M) is composed of 12 layers, 16 attention heads, and an embedding size of 512; the baselines (42M) uses their default configuration with 5 layers, 16 attention heads and embedding size of 512. In Maniskill2, Actra (198M) and the baselines are all composed of 10 layers, 20 attention heads, and an embedding size of 1280. All models are trained using the AdamW optimizer (Loshchilov & Hutter, 2019) with the same hyperparameters within each benchmark, such as the number of epochs, batch size, and learning rate. Benchmark-specific details will be provided in their respective sections.

## 4.2 PERFORMANCE COMPARISON ON VIMA-BENCH

VIMA-Bench (Jiang et al., 2022) focuses on multi-modal robot learning, where the prompts provided to robots are multi-modal. It evaluates generalization capabilities across four levels: placement generalization, combinatorial generalization, novel object generalization, and novel task generalization. Each level presents increasing difficulty, with placement generalization involving only the randomization of object positions, combinatorial generalization recombining seen adjectives and nouns, novel object generalization introducing unseen adjectives and nouns, and novel task generalization incorporating entirely new meta-tasks.

We include the baselines from the original paper and use the same implementation. According to their findings, models with cross-attention and self-attention achieve comparable performance only when the model size exceeds 42M parameters. Therefore, we adopt this configuration for all baselines to strike a balance between model size and performance. We intentionally reduce our model's size by approximately 10% to demonstrate the increased capacity afforded by our optimized architecture. The prompt encoder is T5 (Raffel et al., 2020) and the vision encoder is ViT (Dosovitskiy et al., 2021). Discrete SE(2) actions are used in this benchmark. All models are trained for 10 epochs with learning rate = $1 \times 10^{-4}$ and weight decay = 0.1. Each task is evaluated using 10 trials and the results are summarized in Table 1. The baselines provided in VIMA-Bench explore different methods of encoding images as well as two types of attention mechanisms. We use the best-performing object tokens as visual inputs. Our model demonstrates improved performance across all four generalization levels, despite utilizing only 90% of the baseline model size. Contrastive dynamics learning further boosts performance on the first two levels but lowers it on L3 and L4. This may be attributed to CDL's training data, which only includes seen nouns and adjectives,

Table 2: Performance comparison of success rate (%) on the Maniskill2 benchmark. "Cont." stands for container.

| Model | Configuration | | | Unseen tasks | | | | | Distractors | | | |
|---|---|---|---|---|---|---|---|---|---|---|---|---|
| | Attn Type | Params | Color | Size | Shape | Cont. | All | 0 | 2-4 | 6-8 | Overall |
| RT-1 | — | 46M | 27.03 | 6.36 | 20.30 | 0.79 | 1.27 | 61.09 | 39.17 | 23.40 | 22.43 |
| VIMA | Cross Attn | 525M | 26.00 | 26.00 | 17.20 | 30.75 | 19.33 | 47.93 | 41.47 | 36.33 | 30.63 |
| Gato | Self Attn | 198M | 46.00 | 74.00 | 42.00 | 44.40 | 40.00 | 76.40 | 73.33 | 62.67 | 57.35 |
| Actra (ours) | Traj. attn | 198M | **72.00** | **91.00** | **52.40** | **63.43** | **70.67** | **90.93** | **90.53** | **79.07** | **76.25** |

Table 3: Performance comparison of success rate (%) on the CALVIN benchmark under the most challenging ABC→D setting.

| Model | Configuration | | | Tasks completed in a row | | | | | |
|---|---|---|---|---|---|---|---|---|---|
| | Attn Type | Vision Encoder | Params | 1 | 2 | 3 | 4 | 5 | Avg. Len. |
| MCIL | RNN | ConvNet | 63.6M | 31.0 | 7.9 | 1.4 | 0.0 | 0.0 | 0.40 |
| DT | Self Attn | ViT w/ CLS | 44.1M | 43.5 | 19.4 | 3.2 | 3.2 | 0.0 | 0.69 |
| Gato | Self Attn | ViT w/ Perceiver | 44.1M | 46.0 | 17.5 | 4.8 | 1.6 | 0.0 | 0.70 |
| VIMA | Cross Attn | ViT w/ CLS | 42.4M | 39.2 | 13.6 | 3.6 | 0.7 | 0.2 | 0.57 |
| Flamingo | Cross Attn | ViT w/ Perceiver | 42.4M | 39.2 | 13.2 | 4.3 | 1.0 | 0.2 | 0.59 |
| Actra (ours) | Traj. Attn | ViT w/ Perceiver | 37.8M | **56.5** | **30.6** | **12.9** | **9.7** | **3.2** | **1.13** |

enhancing the model's performance on seen meta-tasks at the expense of generalizability to novel nouns, adjectives, and meta-tasks.

## 4.3 PERFORMANCE COMPARISON ON MANISKILL

In the Maniskill environment (Gu et al., 2023), we evaluate one of the most commonly utilized skills, "pick and place", with everyday objects with complex geometries. Its goal is to pick up a target object and place it into a container. A language prompt specifies which target object and container are intended, with one prompt corresponding to one task. To test generalization capability, we limit the training set to 15 tasks and evaluate the models on 34 tasks. This benchmark spans generalization levels L1 to L3 in VIMA-Bench (Jiang et al., 2022): all items are randomly placed and the robot pose is randomly initialized, thereby including placement generalization; novel target objects are also introduced as unseen tasks, facilitating both combinatorial and novel object generalization. We compare success rates across various types of unseen objects, including unseen target objects, containers, and distractors, as detailed in Table 2. The training data corresponds to the "2-4 Distractors" setting. This benchmark consists of five types of unseen tasks. The first three types involve target objects with unseen colors, sizes, and shapes. For example, the apple is part of the training data, while the bowl, with its novel shape, is not. The fourth type introduces unseen containers. The fifth type composes all of the first four types. A distractor is an item that is neither the target object nor the container. They are randomly sampled from a diverse pool of items. For all five types of "unseen tasks", we randomly sample and place 2-4 distractors. For seen target objects, we explore whether the number of distractors can impact the models' performance.

We maintained a similar model size across all models in VIMA-Bench. However, in this benchmark, we experiment with the same number of Transformer layers for Actra, VIMA, and Gato. We utilize T5 (Raffel et al., 2020) for language prompts and ResNet (He et al., 2016) for images. We also use discrete 6D pose actions in this environment. RT-1 retains its original configuration with 46M parameters. We train the models for 5 epochs with learning rate $= 1 \times 10^{-4}$ and weight decay $= 1 \times 10^{-4}$. We conduct 50 trials for each task, and each trial is limited to 100 timesteps before a timeout. Due to the additional cross-attention layers in VIMA, its model size is significantly larger, which may have contributed to overfitting in this experiment. Our model matches the parameter count of Gato while achieving superior performance and generalization.

## 4.4 PERFORMANCE COMPARISON ON CALVIN

The CALVIN benchmark (Mees et al., 2022) focuses on long-horizon manipulation tasks. During each evaluation session, the model is prompted with five random tasks in a specific order. The session terminates as soon as a task fails, and the remaining tasks are not attempted. Performance is measured by the number of tasks successfully completed in a row. The benchmark provides three

Table 4: Ablation study of the proposed components in Actra. Starting from the row "Actra", we exclude contrastive dynamics learning.

| Config | VIMA-Bench | | | | | Maniskill (Seen, 2-4 Distractors) | | | |
|---|---|---|---|---|---|---|---|---|---|
| | L1 | L2 | L3 | L4 | Overall | Easy | Medium | Hard | Overall |
| Actra w/ CDL | **86.92** | **86.15** | 83.33 | 35.00 | 72.85 | **93.83** | **89.71** | **83.50** | **90.53** |
| Actra | 83.08 | 81.54 | **84.00** | **50.00** | **74.66** | 91.33 | 88.57 | 75.00 | 87.87 |
| w/o Traj Attn | 80.76 | 78.46 | 82.49 | 45.00 | 71.68 | 86.33 | 83.71 | 76.00 | 83.73 |
| w/o Act Query | 61.54 | 54.62 | 57.50 | 25.00 | 49.67 | 86.00 | 80.86 | 69.00 | 81.33 |
| w/o Both | 48.46 | 49.23 | 43.33 | 17.50 | 39.63 | 72.33 | 76.00 | 67.00 | 73.33 |

different experimental settings: D→D, ABCD→D, and ABC→D, where each letter represents a distinct environment. In the D→D setting, the model is both trained and evaluated in environment D. However, in the ABC→D setting, the model is trained on data from environments A, B, and C, but evaluated in environment D. Thus, the ABC→D setting assesses the model's capacity for zero-shot generalization. We compare our model to baselines in this most challenging ABC→D experiment. Since only 1% of the training dataset for ABC→D is annotated with language prompts, we utilize this language-annotated subset for training, further increasing the difficulty of the task.

We use CLIP language encoder (Radford et al., 2021) and MAE-ViT vision encoder (He et al., 2022). Continuous 6D pose actions are used in all models. The models are trained for 10 epochs with learning rate $= 9 \times 10^{-4}$ and weight decay $= 1 \times 10^{-4}$. Performance comparison results are presented in Table 3. Since the rollout of a trajectory terminates as soon as any of the five tasks fails, successfully completing all five tasks is highly challenging. Our model is able to complete longer task sequences than all baselines, highlighting its effectiveness in generalizing to new environments.

### 4.5 ABLATION STUDY

In our ablation study, we evaluate the effects of the proposed approaches, as shown in Table 4. In Maniskill, we provided more granular results across three difficulty levels: hard, medium, and easy (Appendix A.4). Contrastive dynamics learning proves effective in enhancing primary robot imitation learning, particularly for placement and combinatorial generalization. CDL enables the model to better learn environment dynamics. However, since the training data is limited to seen nouns, adjectives, and meta-tasks, this improvement may come at the cost of reduced generalization to novel objects and tasks.

We further investigate the impact of ablating trajectory attention and action query in Actra. The removal of trajectory attention results in a noticeable decrease in success rates across all levels, underscoring its crucial role in processing segmented multi-modal trajectories. Similarly, the absence of action queries leads to reduced success rates, highlighting its importance in enhancing information extraction for action generation. When both components are removed, the model reverts to a typical token-level autoregressive model, akin to Gato Reed et al. (2022).

### 4.6 QUALITATIVE ANALYSIS

We present the loss and accuracy curves for the models on VIMA-Bench in Figure 4. The Flamingo baseline is excluded from the figure due to its significantly worse performance, with both loss and accuracy falling outside the plotted range. Despite having a model size approximately 10% smaller, our model exhibits a much faster convergence rate. The substantially lower loss and higher accuracy explain the superior performance of Actra. This highlights that Actra's architecture enables greater model capacity even with fewer parameters. Specifically, trajectory attention facilitates improved information flow within each segment, while action queries efficiently extract embeddings dedicated to individual action dimensions. The combination of these techniques optimizes performance over previous Transformer architectures.

In Maniskill, we identified a crucial distinction between Actra's capabilities and those of the baselines: Actra masters "instantaneous regrasp". Figure 5 presents key frames from an instantaneous regrasp corresponding to the most challenging task of "pick blue tea box and place into clear box". In most cases, baseline models struggle to recognize failed grasps. Even if they identify a failed grasp, the time taken to start a new attempt is usually prolonged. In contrast, our Actra model

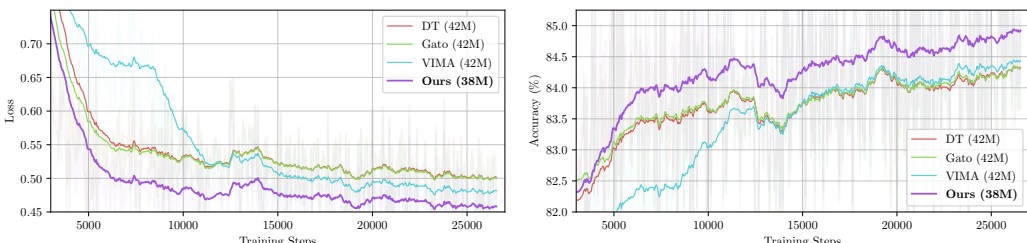

Figure 4: Loss and accuracy curves during training on VIMA-Bench.

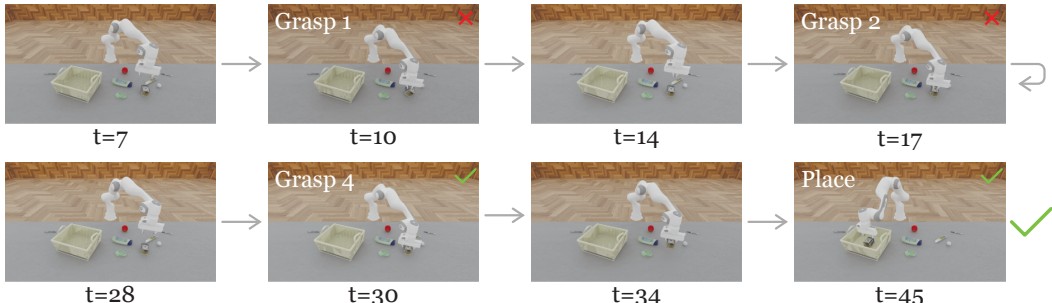

Figure 5: An example of instantaneous regrasp. Four grasp attempts were completed within only 30 steps, a capability not observed in the baseline models. An elaborate description of the example is provided in Appendix A.5.

promptly detects a failed grasp and repeatedly attempts to grasp the object until successful. This ability significantly reduces the failure rate, contributing to our substantially higher success rate.

## 5 CONCLUSION

This paper introduces Actra, an optimized Transformer architecture designed for multi-modal trajectories in robotic tasks. Actra distinguishes itself from vanilla Transformer decoders through two key components: trajectory attention and action query. Trajectory attention harnesses the unique characteristics of multi-modal trajectories, facilitating enhanced information flow among tokens within each segment. This allows Actra to encode the sequence at the segment level, and we introduce action queries to enable a segment-level decoding procedure. We incorporate an additional contrastive dynamics learning objective to explicitly train the model to learn environment dynamics, which also improves multi-modal alignment. This further elevates Actra's performance in robot imitation learning. Through comprehensive comparisons across various benchmarks, our approach demonstrates substantial performance gain over state-of-the-art models. Detailed ablation studies and qualitative analyzes further validate the effectiveness of Actra.

## 6 LIMITATIONS AND FUTURE DIRECTIONS

While more powerful language and vision encoders could be explored for further performance gains, we intentionally refrain from incorporating frontier language or vision models into Actra to ensure a fair comparison with the baselines. Due to the nature of imitation learning, model performance is inherently upper-bounded by the quality of the demonstration data. Unlike NLP, the limited availability and diversity of robot pretraining data restrict the performance gains achievable through contrastive dynamics learning. Moreover, the significant differences across various benchmarks— such as camera settings, action types, and meta-tasks—make it impractical to train a single policy capable of handling all benchmarks. This points to an intriguing future direction: unifying policies across different robotic environments, which may require significantly larger models, such as large language models.

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

# A APPENDIX

## A.1 TRAJECTORY ATTENTION MATRIX

The trajectory attention is implemented with its corresponding attention mask, as illustrated in Figure 6. Output tokens on the left attend to input tokens at the top. For example, in the first row, the token $p_1$ can attend to tokens $(p_1, p_2, p_3, p_4)$ and no future tokens, hence other tokens starting from $s_1$ are masked out. For action generation, action queries can attend to all tokens up to the current state, while no other tokens can attend to action queries. Therefore, action query tokens are all masked from the input.

Although we can use the decoding attention matrix for both decoding and encoding, employing the right matrix can save compute for the action queries. Regardless of using the left or right matrix, the resulting embeddings are identical for the encoded trajectory in VLA contrastive learning, thanks to our modified positional embeddings.

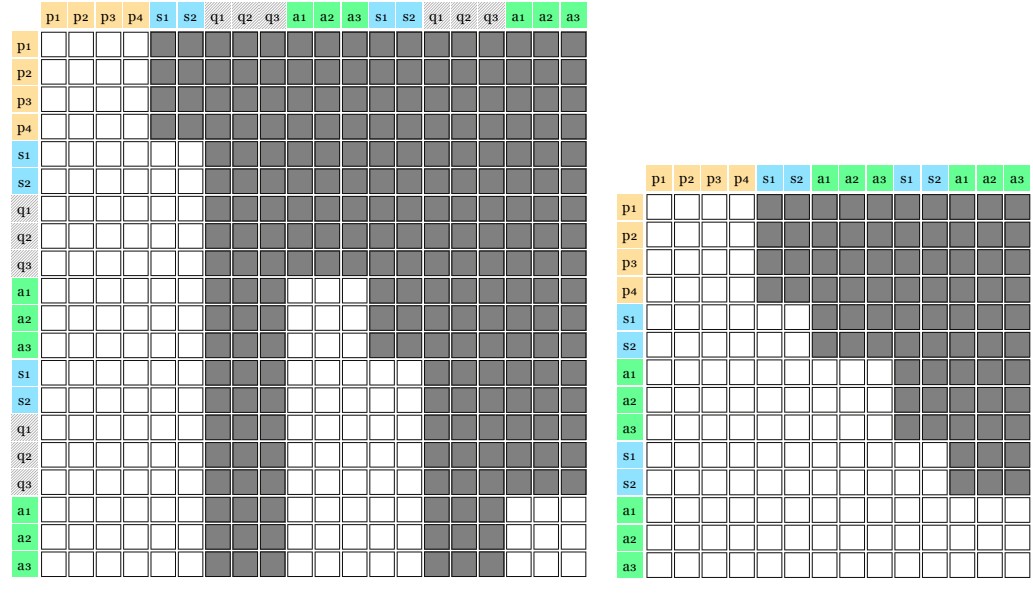

(a) Trajectory attention matrix for decoding.    (b) Trajectory attention matrix for encoding.

Figure 6: The attention matrix of Trajectory Attention. Dark boxes represent masked entries in the attention matrix. The left attention matrix is used for decoding during action generation while the right attention matrix is used for encoding the trajectory in VLA contrastive learning.

## A.2 TRAINING OBJECTIVES OF DIFFUSION-BASED VLA MODELS

The training objectives in diffusion-based VLA models (Ho et al., 2020; Austin et al., 2021) can be written as:

$$\mathcal{L}_{\text{DDPM}} = \text{MSE}\left(\varepsilon^k, \varepsilon_\theta\left(\mathbf{x}^0 + \varepsilon^k, k\right)\right)$$
$$\mathcal{L}_{\text{D3PM}} = \text{CE}\left(\varepsilon^k, \varepsilon_\theta\left(\mathbf{x}^0 + \varepsilon^k, k\right)\right) \tag{2}$$

where $\mathbf{x}^0$ is the original action and $\varepsilon^k$ is the noise of the $k$-th iteration; $\varepsilon_\theta$ is the VLA model.

## A.3 GENERALIZATION LEVELS

For Actra, Gato, and VIMA, unseen shape proves to be the most challenging level, followed by unseen containers. VIMA also exhibits volatility when dealing with small objects from the "Size" level. RT-1, in particular, struggles with identifying the container when an unseen container is introduced. It's important to note that not all seen target objects have a generalized version for every

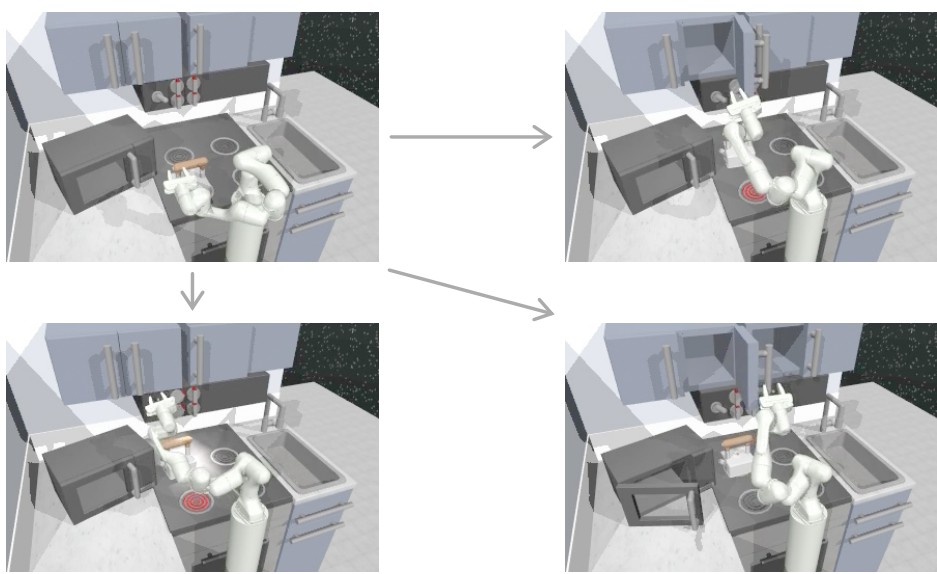

Figure 7: Actra in Franka Kitchen. The model completes four random tasks. The top left image shows the initial state and the other three images are three different final states.

generalization level. For example, a strawberry is a seen target object, but there is no over-sized strawberry for the "Shape" level. Consequently, models may achieve a higher success rate on some generalization levels than on seen tasks. Furthermore, since unseen color and size are part of the mixture, the success rate in "Both" is not as low as in "Shape" and "Container". The introduction of more distractors in the scene increases the likelihood of collisions and causes additional difficulty in grasping the objects. However, this negative effect is not severe enough to considerably degrade the performance.

## A.4 DIFFICULTY LEVELS

In simple terms, easy tasks involve spherical, regular-sized target objects, such as a baseball. The medium difficulty level includes elongated or small target objects, such as a banana or strawberry. Hard tasks encompass oversized, non-spherical, or thin objects, such as a tea box or knife.

Easy tasks include spherical, regular-sized objects. The reason why round objects are easier to pick is that the robot arm can close the gripper in any direction. Size also has a big impact on the success rate because over-sized objects require more precise grasp poses. If a grasp is not precise, the two fingers of the gripper might have collision with the object and not be able to reach down on the object. Small objects can increase the difficulty because the gripper might miss them if the grasp is slightly off. An object like a remote controller or a banana should be picked up "across" the object, not "along" the object. Thus, we define the medium difficulty level as the objects that are too big, too small, or elongated. Hard tasks involve non-spherical, oversized, or thin objects, such as a tea box and a knife. A tea box is non-spherical and oversized and thus the robot arm can only grasp it precisely in parallel with the sides, not diagonally. A knife can be hard since it is very thin and close to the desk. The gripper might collide with the desk while grasping.

## A.5 INSTANTANEOUS REGRASP

More explanation for the example in Figure 5. The first grasp was unsuccessful as one finger of the gripper collided with the blue tea box and the grasp slipped. Subsequently, Actra swiftly initiated two additional grasps; however, the gripper closed too early, resulting in collisions with the tea box again. Shortly after the second and third failures, the gripper's fingers successfully reached down to opposite sides of the blue tea box, completing the fourth regrasp. Remarkably, all four grasp attempts were executed within a mere 30 timesteps, a feat not observed in the baselines.

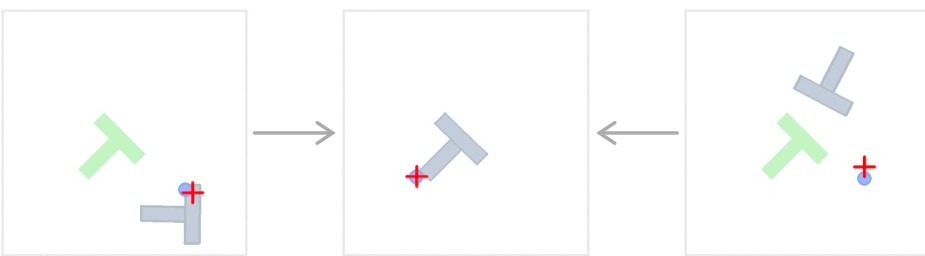

Figure 8: Actra in Push-T. The grey T is the object and the green T is its target position. The model controls the blue dot to push the T-shaped object towards the target position. The red cross is the cursor. The images on the left and right are two different initial states, and the middle image is the final state where the object perfectly overlaps with the target position.

Table 5: Performance comparison (%) in Franka Kitchen and Push-T.

| Model | Franka Kitchen Config | Params | Continuous Action p1 | p2 | p3 | p4 | Mean | Discrete Action p1 | p2 | p3 | p4 | Mean | Push-T Discrete Action Config | Params | Score |
|---|---|---|---|---|---|---|---|---|---|---|---|---|---|---|---|
| Diffusion | base | 43M | **100** | 94 | 72 | 34 | 76 | 54 | 18 | 10 | 2 | 21 | base | 43M | 83.89 |
| VIMA | base | 113M | 92 | 90 | 80 | 66 | 82 | 90 | 72 | 48 | 16 | 57 | small | 26M | 91.09 |
| Gato | base | 43M | **100** | 98 | 84 | 62 | 86 | **100** | 88 | **68** | 38 | 74 | small | 19M | 90.43 |
| Actra | base | 43M | **100** | **100** | **94** | **70** | **91** | **100** | **94** | **68** | **52** | **79** | small | 19M | **94.11** |

## A.6 ADDITIONAL RELATED WORK

MOO (Stone et al., 2023) introduced multi-modal prompt capability to RT-1, while Q-Transformer (Chebotar et al., 2023) adapted RT-1 to the Q-learning setting. RoboFlamingo (Li et al., 2023b) constructed a VLA based on the existing Flamingo VLM (Alayrac et al., 2022; Awadalla et al., 2023). ACT (Zhao et al., 2023) adopts the DETR framework for robotics tasks but utilizes fixed position embeddings at the timestep level. Another category of VLAs focuses on building high-level planners for long-horizon robotics tasks and abstracting away the low-level control policies, such as SayCan (Ichter et al., 2022), PaLM-E (Driess et al., 2023), and ChatGPT for Robotics (Vemprala et al., 2023).

In addition to the primary learning objective, auxiliary or pretraining objectives have proven useful in further enhancing model performance. The success of masked language modeling, as initially proposed in BERT (Devlin et al., 2019), has prompted the adoption of similar objectives in various domains. In computer vision models and VLMs, representative works like MAE (He et al., 2022) and ViLBERT (Lu et al., 2019) have employed comparable strategies. VLAs have also utilized masked modeling objectives for their vision encoders, such as MVP (Radosavovic et al., 2022), Voltron (Karamcheti et al., 2023), GR-1 (Wu et al., 2023). While these approaches have proven beneficial for the vision encoder, they often overlook the crucial alignment between different modalities.

## A.7 ADDITIONAL BENCHMARKS: FRANKA KITCHEN & PUSH-T EXAMPLES

Actra-small (19.4M) consists of 6 layers, 8 attention heads, and an embedding size of 512. Actra-base (43.3M) comprises 6 layers, 12 attention heads, and an embedding size of 768. The baseline models adopt the same configurations unless specified otherwise. We provide some execution examples by Actra in Franka Kitchen (Figure 7) and Push-T (Figure 8).

**Franka Kitchen.** Franka Kitchen (Gupta et al., 2019) includes five skills that span seven specific tasks within the scene. The "turn knob" skill involves turning the oven knob to activate either the top or bottom burner. "Toggle switch" involves turning on the light switch. "Slide door open" requires opening the slide cabinet, while "swing door open" involves opening either the left hinge cabinet or the microwave door by the door handle. The "lift by handle" skill entails moving the kettle by its handle. Performance is measured by the completion of multi-stage tasks, as summarized in Table 5. Results are averaged over 50 runs. In each run, the models are required to complete four random tasks within 280 steps. If $pi = 1$, it means the model has completed $i$ tasks and thus reached the

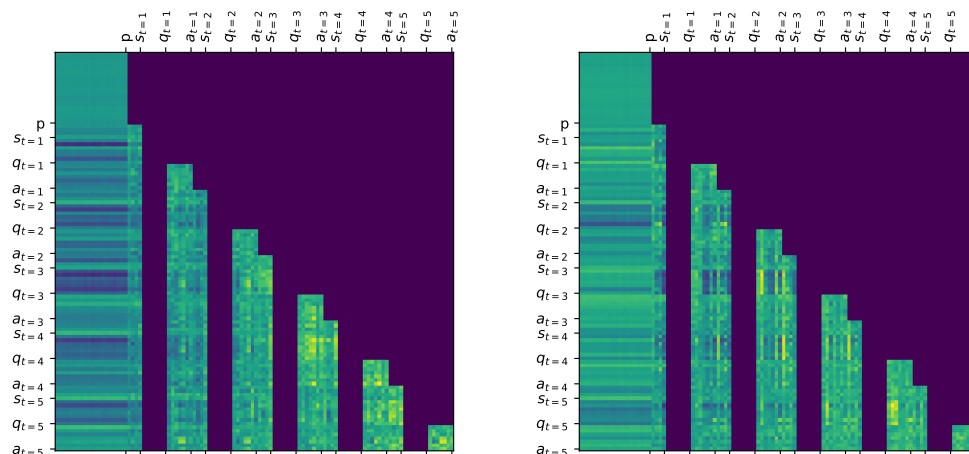

Figure 9: The attention matrix of trajectory attention (left, showing two timesteps) and the visualization of two trained matrices (middle and right, showing five timesteps). Brighter cells correspond to higher attention weights. The tick labels are shown for the last token of every segment.

$i$-th stage; otherwise, $pi = 0$. Since the scale of Franka Kitchen is relatively small, we compare the models in their `base` configuration. We also compare the performance of models using continuous and discrete actions in this environment. Some examples by Actra in Franka Kitchen are shown in Appendix A.7.

**Push-T.** In Push-T (Florence et al., 2021), the models need to push a T-shaped object until it aligns perfectly with the target position. This task requires precise control, as performance is measured by the overlapping area, with perfect alignment equating to a score of 1.0. The performance of the models is compared in Table 5. Results are averaged over 30 trials. The maximum number of steps the models can take in each trial is 200, so they need to push the object precisely while maintaining adequate speed. Because this is a 2D task, we found that using `small` models is sufficient, except for the diffusion-based model, which uses the `base` configuration. Since the cross-attention layers in VIMA make its default `small` model amount to 50.8M parameters, which is even larger than Actra-`base`, we use three Transformer blocks instead. We found that models fail to learn effective policies using continuous actions; therefore, we only report results of discrete actions. Several push-T examples by Actra are included in Appendix A.7.

A.8   TRAJECTORY ATTENTION VISUALIZATION

We present a visualization of the attention matrices from the top layer of Actra, depicted in Figure 9. The explanation of the attention matrix can be found in the appendix. In the visualization, prompt tokens exhibit similar attention values. Notably, tokens from more recent timesteps receive a higher attention weight compared to those from earlier in the sequence. This aligns with our expectation that the latest timestep is the most informative one for generating the next action. Some of the attention weights above the main diagonal are strongly activated, indicating the additional attention connections facilitated by our trajectory attention are beneficial. In the right matrix, a clear distinction is observed between the attention weights produced by state tokens and query tokens. This distinction underscores that action queries extract information differently from state tokens, elucidating their role in improving action generation.

