# OpenReview forum: "Actra: Optimized Transformer Architecture for Vision-Language-Action Models in Robot Learning"
_ICLR.cc/2025/Conference — ICLR 2025 Conference Withdrawn Submission_

### Official Review · Reviewer_7Gym · 2024-11-01

**Soundness:** 3
**Presentation:** 3
**Contribution:** 2
**Rating:** 5
**Confidence:** 4

**Summary:**

Vision language action models typically use causal attention, which may be suboptimal for processing multimodal sequences. The paper makes two contributions: 1) analysis of how the attention pattern of VLAs can be modified to improve performance, and 2) sequence-level contrastive learning helps. Simulation experiments on VIMABench, Maniskill2, and CALVIN show that the method outperforms AR models and shows the effectiveness of contrastive learning.

**Strengths:**

1. The paper explores an interesting direction: whether the autoregressive mask pattern makes sense for VLAs. While it is less well supported by efficient inference methods (i.e., flash-attention, ring-attention, etc.), the reviewer believes that exploring such direction is beneficial for the community.
2. Thorough ablation studies. It helps the reader understand how the change in attention pattern affects the model's performance.

**Weaknesses:**

1. Certain claims in the abstract or introduction may not be correct and need to be elaborated.

“Additionally, the autoregressive generation approach falls short in generating multi-dimensional actions” (Line 15-16)
Many existing VLAs (such as Octo [1] and OpenVLA [2]) are AR models. Many behavior cloning models [4] are also AR models. They all generate multi-dimensional actions and perform real robot tasks.

“VLAs predominantly build upon the pioneering foundations laid by Decision Transformer and Trajectory Transformer … This paradigm has become a cornerstone across recent VLAs” (line 46-49).
These two formulations are drastically different structures than RT-2, OpenVLA, etc, where there’s no interleaving of vision, action, and proprioception data and do not model long sequences. Instead, many recent approaches have gone towards VQA models such as LLaVA / BLIP / Transfusion.

2. The paper can benefit from comparisons with the current state-of-the-art VLAs (Octo [1] and OpenVLA [2]). Additionally, many recent works such as Implicit behavior cloning [5] and diffusion policy [6] can work with <50 demonstrations. Multi-task diffusion-based VLA (Octo, scaling up and distilling down [7]) also shows promise in few-shot learning. It is not obvious that the proposed method is more sample-efficient then these methods.

3. Lack of ablations and experiments on a real robotics setup. 4. The paper does not report confidence intervals (standard deviation or standard error) on the results. It is hard to conclude from the reported results to claim that contrastive learning helps Actra.
5. Scalability: many of the baselines (i.e. RT-1, Gato, VIMA) can benefit significantly from pre-training (in fact most of the evaluations for these models are done post large-scale pre-training). It is not obvious if this method is as scalable (both in terms of model sizes and dataset sizes) compared to the baselines.
6. Past works have explored sequence-level contrastive learning (i.e. ConDT [3]). The paper can benefit from including more ablation studies over different types of sequence-level contrastive learning frameworks.

[1] Team, Octo Model, Dibya Ghosh, Homer Walke, Karl Pertsch, Kevin Black, Oier Mees, Sudeep Dasari et al. "Octo: An open-source generalist robot policy." arXiv preprint arXiv:2405.12213 (2024).

[2] Kim, Moo Jin, Karl Pertsch, Siddharth Karamcheti, Ted Xiao, Ashwin Balakrishna, Suraj Nair, Rafael Rafailov et al. "OpenVLA: An Open-Source Vision-Language-Action Model." arXiv preprint arXiv:2406.09246 (2024).

[3] Konan, Sachin G., Esmaeil Seraj, and Matthew Gombolay. "Contrastive decision transformers." In Conference on Robot Learning, pp. 2159-2169. PMLR, 2023.

[4] Radosavovic, Ilija, Bike Zhang, Baifeng Shi, Jathushan Rajasegaran, Sarthak Kamat, Trevor Darrell, Koushil Sreenath, and Jitendra Malik. "Humanoid locomotion as next token prediction." arXiv preprint arXiv:2402.19469 (2024).

[5] Florence, Pete, Corey Lynch, Andy Zeng, Oscar A. Ramirez, Ayzaan Wahid, Laura Downs, Adrian Wong, Johnny Lee, Igor Mordatch, and Jonathan Tompson. "Implicit behavioral cloning." In Conference on Robot Learning, pp. 158-168. PMLR, 2022.

[6] Chi, Cheng, Zhenjia Xu, Siyuan Feng, Eric Cousineau, Yilun Du, Benjamin Burchfiel, Russ Tedrake, and Shuran Song. "Diffusion policy: Visuomotor policy learning via action diffusion." The International Journal of Robotics Research (2023): 02783649241273668.

[7] Ha, Huy, Pete Florence, and Shuran Song. "Scaling up and distilling down: Language-guided robot skill acquisition." In Conference on Robot Learning, pp. 3766-3777. PMLR, 2023.

**Questions:**

1. Claim in abstract: “Additionally, the autoregressive generation approach falls short in generating multi-dimensional actions.” Can you elaborate why it AR has fall short in action generation?
2. How many tokens are there to represent action at each timestep?
3. How does the sample complexity of the model compare to diffusion policy [1] (for a single task setup, such as that in table 5)?
4. What’s the sequence length of the model? After adding CDL, how much slowdown is present for training?

[1] Chi, Cheng, Zhenjia Xu, Siyuan Feng, Eric Cousineau, Yilun Du, Benjamin Burchfiel, Russ Tedrake, and Shuran Song. "Diffusion policy: Visuomotor policy learning via action diffusion." The International Journal of Robotics Research (2023): 02783649241273668.

---

### Official Review · Reviewer_5hxd · 2024-11-03

**Soundness:** 2
**Presentation:** 3
**Contribution:** 2
**Rating:** 3
**Confidence:** 3

**Summary:**

The paper studies the problem of learning VLA models for robot manipulation tasks. The proposed method optimizes the architecture of classical transformer models by introducing trajectory attention. The authors also propose to use learnable action queries to make the action decoding more focused on relevant information. The method is validated on several simulation benchmarks.

**Strengths:**

The method achieves good performance from the evaluation perspective;

The paper is well structured and easy to follow.

**Weaknesses:**

The design is not sufficiently evaluated through experiments.

The comparison with baselines is insufficient, more prior works need to be included and compared, e.g., Behavior Transformers[1], CoTPC[2].

[1] Behavior Transformers: Cloning k modes with one stone, NeurIPS’22.

[2] Chain-of-Thought Predictive Control, ICML’24.

**Questions:**

How would the proposed attention mechanism affect the inference speed?

What are the failure modes of the method? Providing additional analysis would help readers better understand the work.

---

### Official Review · Reviewer_6Gjy · 2024-11-04

**Soundness:** 2
**Presentation:** 3
**Contribution:** 1
**Rating:** 3
**Confidence:** 4

**Summary:**

The paper proposes optimized Transformer Architectures for Policy Learning. The authors propose 3 contributions: a custom self-attention mask for policy learning, learnable action queries for each action dimension and an contrastive loss function to improve policy learning by having the model differentiate between plausible dynamics and unrealistic ones. The proposed architecture is tested on three common IL benchmarks including CALVIN and VIMA with several ablations.

**Strengths:**

- novel dynamics self-supervised loss to improve policy learning performance, which does not require pretraining and improves performance

- proposed architecture changes improve upon a wide range of transformer baselines in different benchmarks

- the paper is overall well written and easy to follow with several illustrative figures

**Weaknesses:**

- The terminology "VLA" is misleading since the model lacks vision-language pretraining. This is fundamentally a standard Transformer-based policy without language generation or image understanding capabilities.

- Performance issues:

    - CALVIN results are significantly below SOTA (1.x vs 3-4 average rollout length from methods like GR-1 or 3DDA), with existing methods like GR-1 already implementing similar trajectory attention mechanisms and learnable action queries
    - VIMA improvements are marginal, with lower performance compared to recent approaches like [4] that also utilize per-dimension action tokens
    - Low Diffusion baseline performance on Relay Kitchen and Push-T compared to originally reported metrics - what explains that discrepancy?
- Limited novelty in technical contributions:
    - Trajectory attention is an incremental modification seen in multiple prior works (Octo [1], GR-1 [2])
    - Learnable action queries are also incremental and were previously explored in GR-1, ACT [3] or [4] and similar research
    - No substantial architectural innovations except the self-supervised loss

- Methodological limitations:
    - Absence of scaling analysis across model sizes and environments
    - No large-scale experimental validation
    - Training duration of 5-10 epochs is insufficient to demonstrate convergence
    - Lacks comprehensive ablation studies on larger-scale settings
    - Design changes of generalist policies need evaluation on large-scale datasets to see if the proposed changes scale as intended


Sources:

[1]: Team, Octo Model, et al. "Octo: An open-source generalist robot policy." RSS 2024

[2]: Wu, Hongtao, et al. "Unleashing large-scale video generative pre-training for visual robot manipulation." ICLR 2024

[3]: Zhao, Tony Z., et al. "Learning fine-grained bimanual manipulation with low-cost hardware." RSS 2023

[4]: Li, Jiachen, et al. "Mastering robot manipulation with multimodal prompts through pretraining and multi-task fine-tuning." _arXiv preprint arXiv:2310.09676_ (2023).

**Questions:**

- Could you elaborate on the low performance of Actra on CALVIN compared to recent sota methods? What factors do you think contributes to these differences and how would you bridge them in future work?

- The training duration of 5-10 epochs shows promising initial results. Have you explored longer training runs to understand the full convergence behavior and potential performance ceiling of the approach?

- How are the contributions of trajectory attention and learnable action queries different to prior work that employed the same already?

- Could you consider including additional scaling experiments to demonstrate how this approach performs across different model capacities and dataset sizes? This could help better understand its broader applicability as a VLM policy.

---

### Note · Authors · 2024-11-27

I have read and agree with the venue's withdrawal policy on behalf of myself and my co-authors.